# Novel Imine-Tethering Cationic Surfactants: Synthesis, Surface Activity, and Investigation of the Corrosion Mitigation Impact on Carbon Steel in Acidic Chloride Medium via Various Techniques

**DOI:** 10.3390/molecules28114540

**Published:** 2023-06-03

**Authors:** Hany M. Abd El-Lateef, Ahmed H. Tantawy, Kamal A. Soliman, Salah Eid, Mohamed A. Abo-Riya

**Affiliations:** 1Chemistry Department, College of Science, King Faisal University, Al-Ahsa 31982, Saudi Arabia; 2Chemistry Department, Faculty of Science, Sohag University, Sohag 82534, Egypt; 3Chemistry Department, Faculty of Science, Benha University, Benha 13518, Egypt; kamal.soliman@fsc.bu.edu.eg (K.A.S.); eedsalah@yahoo.com (S.E.); mohamed.aborya@fsc.bu.edu.eg (M.A.A.-R.); 4Chemistry Department, College of Science and Arts, Jouf University, Alqurayat 77455, Saudi Arabia

**Keywords:** imine-tethering cationic surfactants, surface-active properties, corrosion inhibition, carbon steel, surface morphology, MC simulation

## Abstract

Novel imine-tethering cationic surfactants, namely (E)-3-((2-chlorobenzylidene)amino)-*N*-(2-(decyloxy)-2-oxoethyl)-*N,N*-dimethylpropan-1-aminium chloride (ICS-10) and (E)-3-((2-chlorobenzylidene)amino)-*N,N*-dimethyl-*N*-(2-oxo-2-(tetradecyloxy)ethyl)propan-1-aminium chloride (ICS-14), were synthesized, and the chemical structures were elucidated by various spectroscopic approaches. The surface properties of the target-prepared imine-tethering cationic surfactants were investigated. The effects of both synthesized imine surfactants on carbon steel corrosion in a 1.0 M HCl solution were investigated by weight loss (WL), potentiodynamic polarization (PDP), and scanning electron microscopy (SEM) methods. The outcomes show that the inhibition effectiveness rises with raising the concentration and diminishes with raising the temperature. The inhibition efficiency of 91.53 and 94.58 % were attained in the presence of the optimum concentration of 0.5 mM of ICS-10 and ICS-14, respectively. The activation energy (*E_a_*) and heat of adsorption (*Q_ads_*) were calculated and explained. Additionally, the synthesized compounds were investigated using density functional theory (DFT). Monte Carlo (MC) simulation was utilized to understand the mechanism of adsorption of inhibitors on the Fe (110) surface.

## 1. Introduction

Carbon steel (C-Steel) is the most popular and common construction material across a lot of industries. Carbon steel suffers from serious corrosion problems [1,2,3]. Inhibitors are a common and effective way to safeguard metals in acidic media. [4]. Acidic solutions, particularly HCl acid solutions, are frequently utilized in chemical and industrial operations [5]. The majority of organic molecule inhibitors function by adhering to the metal’s surface and binding to heteroatoms, such as sulfur, oxygen, and nitrogen, as well as many linkages. [6,7]. The surface charge and nature of metals, the chemical construction of organic inhibitors, and the type of aggressive solution all influence this process [8]. Numerous studies have been conducted on the corrosion and inhibition of C-steel in acidic solutions [9,10,11,12].

Cationic surfactants, mainly quaternary amines, are considered one of the most important surfactants having numerous uses in different fields. They are frequently used in a variety of industrial applications, including medicines, pharmaceutical formulations, food processing, and oil recovery [13,14,15]. In recent decades, cationic surfactants have shown promising results as corrosion inhibitors [16,17,18]. This is accredited to the absorption of cationic surfactants on the steel interface, forming an adsorption film layer, which separates the metal surface from the surrounding environment [19]. The efficiency of this material as an inhibitor depends on its adsorption capacity [20]. Most cationic surfactants can be used as corrosion inhibitors containing O, N, S, P, heteroatoms, and/or π-bonds that permit the adsorption of the surfactant molecule on the electrode interface [21,22]. The inhibition process is achieved by the buildup of a coordinate covalent bond between the free electrons of these atoms or multiple bonds and lower unoccupied orbitals on the metal [23,24]. 

Therefore, compounds with these atoms or bonds have a strong ability to resist corrosion. Hence, began an investigation of Schiff base compounds, which contain a heteroatom and π-bond as corrosion inhibitors. Organic substances known as Schiff bases have numerous uses in biology and materials research [25,26]. Schiff-based cationic surfactants can adsorb onto metal surfaces by charge-transfer complex bonds between the polar groups of Schiff base and metal surfaces [27,28]. Cationic surfactants bearing Schiff base moiety proved high efficiency in an acidic solution, where a nitrogen atom is accomplished to form coordinate covalent bonds with the steel interface by its unshared, lone pairs of electrons, whereas the π-bond interacts physically, increasing their adsorption attraction to the metal interface. Imine surfactant molecules are categorized by their high capability to adsorb at the interfaces, owing to their unique amphipathic construction, which contains two opposing parts, a tail, and a head. The head is considered by its high electronic richness, which is a set of some electronic-rich functional groups, such as double, carbonyl, nitrogen, the imine group, oxygen, etc. A successive film on the metal surface could be shaped by the surfactant’s hydrocarbon, which has the capability to separate the corroded metal surface from contact with corrosive mediums [29,30,31,32,33,34]. In addition, the imine cationic surfactant is considered by its biocidal activity, which improves its capacity in the petroleum sector. Furthermore, in acidic conditions, these compounds exhibit effective anticorrosive properties.

The objective of our study was to prepare and study the inhibition effect of (E)-3-((2-chlorobenzylidene)amino)-*N*-(2-(decyloxy)-2-oxoethyl)-*N,N*-dimethylpropan-1-aminium chloride (ICS-10) and (E)-3-((2-chlorobenzylidene)amino)-*N,N*-dimethyl-*N*-(2-oxo-2-(tetradecyloxy)ethyl)propan-1-aminium chloride (ICS-14) on the corrosion of carbon steel in an acidic chloride medium, utilizing the PDP, WL, and SEM methods. In addition, activation energy (*E_a_*) and the heat of adsorption (*Q_ads_*) were computed and explained. In addition, the two inhibitors, ICS-10 and ICS-14, were studied by DFT and MC simulation to understand the corrosion inhibition mechanism. 

## 2. Results and Discussion

### 2.1. Synthesis

As shown in Figure 1, the overall synthesis procedure of the target imine-tethering cationic surfactants consists of two steps. Initially, 2-chlorobenzaldehyde was reserved to react with 3-(*N,N* Dimethylamino)-1-propylamine to form 3-((2-chlorobenzylidene)amino)-*N,N*-dimethylpropan-1-amine (1). The prepared Schiff base (1) was refluxed with decyl 2-chloroacetate and tetradecyl 2-chloroacetate in ethyl acetate as a solvent at 70 °C to produce novel cationic imine surfactants [ICS-10] and [ICS-14] with yields equal to 77 and 81%, respectively. The investigations were conducted on the structure of the synthetic surfactants using various spectroscopic tools, such as IR and NMR spectra. The recorded data in the spectra elucidated the functional groups in the obtained compounds, as recorded in the experimental section. 

### 2.2. Surface Activity Measurements

The surface tension (*γ*) of the imine surfactant solutions vs. their bulk concentrations, in mol/L at 25 °C, were plotted, and the CMC (Critical Micelle Concentration) values were graphically acquired and recorded at the breakpoint of a definite concentration of the prepared surfactants, as shown in Figure 1A. Table 1 lists the CMC values and the surface tension at the CMC (*γ*_CMC_). It could be proven that the generated cationic surfactant’s CMC value decreases as the hydrophobic chain lengthens; this could be related to the growing hydrophobicity and a reduction of the dissolving of the imine surfactants, so the system’s freedom energy increases, which results in a surfactant molecule aggregated into the micelles. In order to avoid contact with the aqueous medium, the hydrophilic group directs to the solution, while the hydrophobic series is towards the interior body of the system, lowering the freedom of the system. So, by the length of the hydrophobic tail, the inclination of the surfactant compound to produce a micelle is, therefore, reduced by the CMC. As depicted in Table 1, the values of the CMC, determined in 1.0 M of HCl (Figure 1B), were partially reduced compared with the values in pure aqueous solutions (Table 1).

Based on the electrical conductivity measurements (K) of the as-prepared surfactants, the CMC values and the dissociation degree of the counter ion (*α*) were evaluated and determined at a certain temperature (298 K), as shown in Figure 1C, where the *α* values attained from the ratio of the slopes above and lower the break, revealing the CMC, were determined. Usually, the degrees of counterion binding (*β*) values are determined according to the following recorded relationship: *β* = 1 − *α* [35,36]. The values of *β* and *α* are recorded in Table 1, which shows that the *β* values reduce and the *α* values rise with a lengthening chain of hydrophobic alkyl at 25 °C [37]. It was noted that the prepared cationic surfactants with higher hydrophobic chain tails had a low CMC value. Furthermore, the CMC values were determined via electrical conductivity, which is the same as those obtained using surface tension.

From the calculated values of *γ*, the effectiveness values of the imine compounds were carried out via the next equation [38]:*Π*_CMC_ = *γ*_0_ − *γ*_CMC_
(1)
where *γ*_CMC_ and *γ*_0_ are the *γ* values at the CMC and pure H_2_O, respectively. From the results shown in Table 1, it was detected that the most effective imine surfactant was ICS-14 which revealed the highest decrease in the *γ* at the CMC, whereas the highest lessening of the *γ* at the CMC reached 37.54 [39].

In addition, the surface area of Amin refers to the packaging density of the as-prepared cationic surfactants at the interface of air/water; this is very significant for explaining the surface features of the imine surfactants. The Γ_max_ value, showing the efficacy of the interfacial adsorption of the surfactant adsorption values, was investigated by the Gibbs adsorption equation [38]: (2)$$\Gamma_{\max}=\frac{-1}{2.303nRT}{\left(\frac{d\gamma }{d\log C}\right)}_{T}$$
where *T* = 298 K, *R* denotes the gas constant, *dγ*/*d*log*C* characterizes the slop, and n symbolizes the number of ions that generate inside the solution via the surfactant molecule dissociation (equal to two in our calculations). The Γ_max_ values are listed in Table 1, where it is exhibited that the lengthening of the hydrophobic chain of the imine surfactants shifts Γ_max_ to lower values of concentrations. Additionally, the *A*_min_ calculations, with the minimum surface area in nm^2^, were carried out via the below equation [40]:(3)$$A_{\min}=\frac{10^{14}}{N_{A}\times \Gamma_{\max}}$$
where *N*_A_ equals 6.022 × 10^23^. As shown in Table 1, the Γ_max_ and *A*_min_ values are listed. The data showed that the *A*_min_ values were raised by lengthening the hydrophobic chain, owing to increasing the area of the hydrophobic chain employed by each inhibitor molecule upsurge; consequently, the Amin was augmented, and the Γ_max_ decreased, whereby in decreasing the Γ_max_, the distances among the surfactant compound upsurged, leading to upsurged Amin values.

According to Gibb’s adsorption equations, the calculation of the thermodynamic parameters of the adsorption and micellization of the imine surfactants (ICS-10 and ICS-14) was conducted, as shown below [41]:(4)$$\Delta G_{mic}^{o}=2.303(2-\alpha )RT\log CMC$$
(5)$$\Delta G_{ads}^{o}=\Delta G_{mic}^{o}-0.0602\pi_{CMC}A_{\min}$$
where $\Delta G_{mic}^{o}$ and $\Delta G_{ads}^{o}$ are the micellization and adsorption-free energies, respectively. As seen in Table 1, the values of $\Delta G_{mic}^{o}$ and $\Delta G_{ads}^{o}$ are continuously negative, indicating the spontaneousness of these two routes, but there are higher increases in the negativity values of $\Delta G_{ads}^{o}$ (via the lengthening of the hydrophobic moiety) than in $\Delta G_{mic}^{o}$, referring to the inclination of the surfactants to be adsorbed at the interface. By inspection of the data in Table 1, it was found that the values of the $\Delta G_{mic}^{o}$ and $\Delta G_{ads}^{o}$ increase by the lengthening of a hydrophobic chain, indicating that the tendency of the molecules is absorbed on the interface.

### 2.3. Weight Loss (WL) Studies

The C-steel corrosion in molar hydrochloric acid in the absence and presence of diverse concentrations of imine surfactants was investigated by WL studies. The effects of the addition of various concentrations of ICS-10 and ICS-14 on the weight loss (A) and inhibition capacity (B) of carbon steel in a 1.0 M HCl solution are presented in Figure 2A,B, and the values are recorded in Appendix A. From Figure 2, it is observed that the rates of corrosion (weight loss) declined with the presence of the imine surfactants, as compared to the blank HCl (the free inhibitor). The IE rises with the increasing surfactant concentration, displaying a maximum rise in the IE of 91.13 and 94.16% at surfactant doses of 0.5 mmol of ICS-10 and ICS-14, respectively. The protection of C-steel corrosion with the presence of imine surfactant molecules can be ascribed to the adsorption of the surfactant onto metallic substrates, which prevents steel contact with the corrosive medium and, consequently, does not allow the corrosion process to occur. The rise in the IE with the increasing surfactant concentration designates that more inhibitor species are adsorbed on the metal interface, leading to the development of a defensive layer at the steel–electrolyte interface [42]. As shown in Appendix A, the inhibition powers of our synthesized imine surfactants (~95) are close to or slightly higher than that of the previous synthetic Schiff base compounds (~94) [34,43,44,45]. Thus, in spite of the yield products’ accounts, the use of synthesized compounds, such as our synthesized cationic surfactants, for the protection of the metal surfaces in the corrosive solutions containing chloride produces more yield than the previous synthetic compounds containing the hydroxyl group [44].

### 2.4. Potentiodynamic Polarization (PDP)

PDP plots for the C-steel corrosion both in the blank and inhibited solutions containing various concentrations of ICS-10 and ICS-14 compounds in a 1.0 M HCl medium are depicted in Figure 3.

It was detected that, from Figure 3, all the diagrams display comparable profiles, which verifies that neither the anodic (viz. metal corrosion) nor cathodic (viz. the reaction of hydrogen evolution, RHE) mechanisms were altered by the addition of the inhibitor to the aggressive medium [46]. From the extrapolation of Tafel branches, numerous electrochemical parameters were calculated; cathodic (*β*_c_) and anodic (*β*_a_) Tafel slopes, the corrosion current density (*i*_cor_), and the potential corrosion (*E*_cor_), which is mentioned in Table 2. The inhibition effectiveness (*IE*/%) and surface coverage (*θ*) were estimated utilizing the next equation [46].
(6)$$IE/\%=\left(1-\frac{i_{surf}}{i_{free}}\right)\times 100=\theta \times 100$$
where *i_free_* and *i_surf_* are the *i*_cor_ without and with the ICS-10 and ICS-14 surfactants, respectively.

Examining Figure 3 exposes that there is no substantial modification in the values of either the *β*_c_ and *β*_a_ upon adding different ICS-10 and ICS-14 amounts, demonstrating that the mechanism of inhibition by the investigated imine surfactants continues via the hindrance of the efficient cathodic and anodic locations on the metal interface [47]. The inhibitory influence of ICS-10 and ICS-14 is correspondingly shown as the current density decline in both the cathodic and anodic lines, which agrees well with previously predicted thermodynamic and surface-active characteristics.

Table 2 outlines the protection capacities’ rise with the increasing ICS-10 and ICS-14 concentration, accomplishing ca. 91.53 and 94.58% in the presence of 0.5 mM [ICS-10 and ICS-14] and agreeing well with the gravimetric investigation findings. Such performance could be reasonable by the rising adsorption of ICS-10 and ICS-14 molecules onto the steel–HCl solution interface [47,48], which is reinforced by increasing a part of the surface coverage (cf. with Table 2). The inhibitor is classified as an anodic or cathodic type if the change of the *E*_cor_ in the presence of the surfactant inhibitor is ±85 mV from that in the absence of the inhibitor [48]. The presence of the ICS-10 and ICS-14 surfactants shifts the *E*_cor_ to values less than 85 mV compared to the blank system (the free inhibitor), indicating that ICS-10 and ICS-14 surfactants can be characterized as mixed-type inhibitors, i.e., supporting the retardation of both the cathodic hydrogen evolution and anodic dissolution of the C-steel reactions [48]. 

### 2.5. Morphology Studies

The surface morphologies determined the severity of the corrosion attack. The SEM observation of the C-steel samples after and before immersion in HCl in the absence and presence of 0. 5 mM of ICS-10 and ICS-14 are presented in Figure 4. In the absence of inhibitors, the morphology of the C-steel samples in the molar HCl revealed that the surface was severely degraded. The surface of the C-steel became smooth in the presence of ICS-10 and ICS-14. These findings show that ICS-10 and ICS-14 adsorbed on the C-steel surface and produced a layer that successfully protected the carbon steel surface from the corrosive ions [10].

### 2.6. Adsorption Isotherm Considerations

Weight loss and PDP measurements have established that imine surfactant molecules could efficiently impede the C-steel corrosion in an acidic chloride solution. Out of data fitting for identifying the best adsorption isotherm model and then selecting the approaching value of *R*^2^, the adsorption route mechanism of a surfactant on steel could be determined as well. The surfactant concentration (*C_inh_*) and the surface coverage (*θ*) were utilized to compute the adsorption equilibrium constant (*K_ads_*), and minor alterations in the coverage could affect the protective efficacy.

The values of *θ* for differing ICS-10 and ICS-14 surfactant concentrations were estimated by using the weight loss results to select the optimal isotherm of adsorption for these compounds. Different models of adsorption isotherms, for example, the Langmuir, Temkin, Freundlich, and Frumkin models, were utilized to fit the weight loss data to comprehend how the surfactant compounds are adsorbed on the steel interface. The closest match was for the Langmuir adsorption isotherm, based on the next equation [49]:(7)$$\frac{C_{surf}}{\theta }=\frac{1}{K_{ads}}+C_{surf}$$
where *K_ads_* represents the adsorption equilibrium constant, and *C_surf_* is the molar dose of the ICS-10 and ICS-14 surfactants. The plots of *C_surf_/θ* vs. *C_surf_* for steel corrosion in 1.0 M of HCl are illustrated in Figure 5. The attained diagrams are linear with higher than 0.999. The intercept allows for the computing of the *K_ads_*. The equilibrium constant is equal to 164 × 10^3^ and 238 × 10^3^ for ICS-10 and ICS-14, respectively. The higher values of the *K_ads_* signify an interaction between the steel interface and surfactant, showing that the surfactant additives are powerfully adsorbed on the electrode substrate and, in turn, deliver superior protection efficacy to the surfactant. 

The following equation describes the relationship between the adsorption’s standard free energy and *K_ads_*, the adsorption equilibrium constant [50]:(8)$$K_{ads}=\left(\frac{1}{55.5}\right)\exp\left(\frac{-\Delta G_{ads}^{o}}{RT}\right)$$
where the molar concentration of H_2_O is represented by the number (55.5), *R* is the gas constant (8.314 J. mol^−1^.K^−1^), and *T* is the absolute temperature. 

The $\Delta G_{ads}^{o}$ values for the ICS-10 and ICS-14 adsorbed on the surface of the C-steel in the molar HCl are equal to −39.0 kJ mol^−1^ and −39.9 kJ mol^−1^. The negative value of $\Delta G_{ads}^{o}$ indicates the spontaneous of adsorption the ICS-10 and ICS-14 on the C-steel surface [10].

### 2.7. Thermodynamic/Adsorption Calculations

Table 3 depicts the temperature effect on the corrosion of C-steel in 1.0 M of HCl in the absence and presence of 0.0005 M of ICS-10 and ICS-14 after 24 h. The data reveal that ICS-10 and ICS-14 compounds are efficient inhibitors. As the temperature increase has an opposite connection with the inhibition effectiveness, the increase in temperature likely leads to the desorption of adsorbed ICS-10 and ICS-14 molecules from the metal interface.

The obvious activation energy (*E_a_*) for the C-steel corrosion in molar hydrochloric acid in the absence and presence of 0.0005 M ICS-10 and ICS-14 was computed using an Arrhenius-type equation [51]:(9)$$\log\left(\frac{CR_{2}}{CR_{1}}\right)=\left(\frac{E_{a}}{2.303R}\right)\left(\frac{1}{T_{1}}-\frac{1}{T_{2}}\right)$$
where *R* symbolizes the universal gas constant, a signifies the Arrhenius pre-exponential factor, *T* is the Kelvin temperature, and *CR*_1_ and *CR*_2_ are the corrosion rates computed after 24 h at temperatures *T*_1_ and *T*_2_, respectively. The values of the activation energies (*E_a_*) were calculated and are given in Table 3. This value in the inhibited system is higher than the value obtained for the blank, which designates that ICS-10 and ICS-14 molecules are adsorbed on the C-steel surface, leading to this increase in the activation energy [52].

The adsorption heat (*Q_ads_*) of the inhibitors was calculated using the next equation [53]:(10)$$Q_{ads}=2.303R\left[\log\left(\frac{\theta_{2}}{1-\theta_{2}}\right)-\log\left(\frac{\theta_{1}}{1-\theta_{1}}\right)\right]\times \left(\frac{T_{1}T_{2}}{T_{2}-T_{1}}\right)$$
where *θ*_1_ and *θ*_2_ are the parts of the covered surface at temperatures *T*_1_ and *T*_2_, respectively. The *Q_ads_* value was computed and is listed in Table 3. It has a negative value, so the inhibitor’s adsorption at the metal–acid interface is exothermic, and the amount of surface coverage is reduced with the temperature rise [54,55].

### 2.8. DFT Calculations

The optimized structure of the two inhibitors, ICS-10 and ICS-14, at the B3LYP level of theory and their frontier molecular orbitals are shown in Figure 6. The binding of the surfactant molecules to the steel interface increases with a higher HOMO energy value, which indicates the greater electron donation of the surfactant molecule to the empty d-orbital of the steel and a lower LUMO energy level, which refers to the ability of the surfactant to gain electrons from the d-orbital of the steel [56]. As seen in Table 4, if we compare the inhibitors in the gas phase, ICS-14 has a higher HOMO and lower LUMO energy value, indicating that ICS-14 shows higher inhibition efficiency than the ICS-10 inhibitor molecule. The adsorption ability of the inhibitor increases with a smaller energy gap. The smaller energy gap refers to higher inhibition efficiency and chemical reactivity [6,57]. As seen in Table 4, ICS-14 has a lower energy gap than the ICS-10 inhibitor molecule.

The HOMO electron distribution for the ICS-10 and ICS-14, as seen in Figure 6, were localized on a chloride ion, while the LUMO electron density was distributed on a phenylmethanimine moiety.

The dipole moment (*µ*) is an important quantum descriptor that reflects the global polarity of a molecule. The *µ* is associated with protection efficacy. The protection efficacy upsurges with the increment of the µ. As shown in Table 4, the ICS-14 inhibitor molecule has the highest dipole moment.

The electrons transferred number (Δ*N*) is well-intended. If the Δ*N* values are ˂3.6, the protection efficacy upsurges by increasing the capability of the surfactant molecules to contribute electrons to the steel interface [58,59]. As seen in Table 4, the ΔN values for the two compounds (ICS-10 and ICS-14) are positive, and the ICS-14 inhibitor shows a higher value than the ICS-10 inhibitor molecule, which indicates that the ICS-14 inhibitor has higher inhibition efficiency than the ICS-10 inhibitor.

The global softness (*σ*) and hardness (*η*) for the two ICS-10 and ICS-14 inhibitors were calculated, which determines the reactivity of the surfactant molecule. The inhibitor with a greater softness value and a lower hardness value is predictable as having the highest inhibition efficiency. Therefore, as seen in Table 4, the ICS-14 inhibitor shows higher inhibition efficiency than the ICS-10 surfactant molecule. The molecular electrostatic potential map (ESP) is a visual tool applied to determine the reactive places of the surfactant molecule. The red and yellow regions refer to a negative ESP. The more negatively charged atoms are, the more reactive the atoms are. As seen in Figure 6, the negative ESP is located on the chloride ion, nitrogen, and oxygen atoms of both inhibitors.

### 2.9. MC Simulation and the Mechanism of Inhibition

MC simulation can determine the most configurations of the surfactant molecules adsorbed on a steel interface. The two studied inhibitors, ICS-10 and ICS-14, were simulated in vacuum and gas phases. As seen in Figure 7, the two inhibitors adsorbed on the Fe (110) in a parallel adsorption configuration, and the adsorption energy values of the surfactants on the Fe (110) were found to be −180.22 and −208.84 kcal/mol for ICS-10 and ICS-14, respectively. The higher adsorption energy of the CS-14 inhibitor than the CS-10 inhibitor leads to the strong interaction of the CS-14 inhibitor on the Fe (110), leading to the production of a film that protects the C-steel surface against the aggressive environment (the HCl solution), leading to higher inhibition effectiveness.

It is widely recognized that the polar units found in polar carbon-based molecules operate as reaction positions for the molecules and speed up the process of adsorption by forming bonds between the polar atoms of the steel surface inhibitors. On the basis of the inhibitor’s orientation, shape, size, and electrical charge, the compound’s efficiency is mostly governed by the degree of adsorption [60,61]. Frequent factors that affect the protective effectiveness of carbon-based corrosion additives are the length of the carbon chain, size, and chemical composition of the organic inhibitor; the conjugated bonding and/or aromaticity in the compound; the type, nature, and some bonding groups or atoms in the compound; charges produced on the surface of the metal; the capability of a film to be multiple or single to procedure cross-linked or dense, and the ability to produce a complex with an efficient inhibitor atom with a metallic interface [62,63]. Taking into account the existing case, according to the presence of the π-electrons of the benzene ring, the electronegative N atom in its structure, and a double bond (-C=C-) (Figure 1), the cationic imine-based surfactant may sensibly substantiate its great protection capacity and usage as an efficient corrosion additive. The π-electrons in the surfactant could not only localize the vacant d-orbital of the steel interface but also may receive the d-orbital electrons of the steel substrate to produce feedback of the steel–surfactant bond. The adsorption of the CS-14 inhibitor over the CS-10 inhibitor results in a strong interaction between the CS-14 inhibitor and C-steel, generating a layer that shields the C-steel surface against the aggressive environment.

## 3. Experimental

### 3.1. Materials

The metal used was carbon steel (C-Steel), which had the following structure (wt./wt.%): 0.05 Ni, 0.02 Cr, 0.0256 Si, 1.81 Mn, 0.09 P, 0.1 C, 0.001 V, 0.01 Mo, 0.03 Cu, and the remainder was iron. 

Furthermore, 3-(*N,N* Dimethylamino)-1-propylamine (99%), 2-Chlorobenzaldehyde (97%), and Decyl (98%) were purchased from Acros Organics Company (Geel, Belgium). Hexadecyl (97%) alcohol was attained from M/s S.D. Fine Chemicals Pvt. Ltd. (Mumbai, India) and tetrahydrofuran (99%) were obtained from Alnasr-Chemical Company. Solvents (ethyl acetate, ethyl alcohol absolute (99%), and diethyl ether (99%)) were obtained from Algomhoria Chemical Co., Cairo, Egypt. All the utilized solvents and reagents were received without further purification.

### 3.2. Equipment and Instrumentation

The construction of the as-prepared imine-tethering cationic surfactants was chemically clarified using melting points (Gallen-Kamp, Cambridge, UK) and FTIR spectrum, performed in potassium bromide on (iS10-FTIR spectrophotometer, Paisley, UK) a Thermo-Nicolet, ^1^H, and ^13^C NMR spectra, and estimated in *d*_6_-DMSO using ALPHA-FT-IR-BRUKER-Pt-ATR. Using the ring method, a Tensiometer-K6 processor (KRÜSS-Company, Hamburg, Germany) was utilized to determine the surface tension of as-prepared cationic surfactants at 25 °C [64] in distilled water and 1.0 M of HCl. The electrical conductivity (K) of synthesized surfactants was identified by an electrical conductivity meter, AD3000 type (EC/TDS), and at a temperature of 25 °C.

### 3.3. Synthesis of the As-Prepared Cationic Surfactant 

An amount of 10 mmol of 2-chlorobenzaldehyde was dissolved in 100 mL of ethyl alcohol and added in a single-neck flask, then mixed with 10 mmol of 3-(*N*,*N*-dimethylamino)-1-propyl amine. The solution mixture was refluxed for 7–8 h at 85 °C via thin-layer chromatography (TLC); the accomplishment of the reaction was detected, and the reaction was left to cool down at room temperature. A pale-yellow solid was obtained and purified by recrystallization in methanol to generate the target compound (**1**) with a yield reach of 90%. The purified compound was utilized directly in the next step.

Compound (**1**) (20 mmol) was dissolved in 100 mL of ethyl acetate and taken in two portions separately to react with 10 mmol of decyl- and tetradecyl-2-chloroacetates, stirred at 70 °C for 30 h in ethyl acetate as a solvent. The solid products were recrystallized from diethyl ether and dehydrated under a vacuum at 40 °C to give the target cationic imine surfactants (ICS-10 and ICS-14). The purification process was completed by removing the undesirable materials during washing with diethyl ether. The structure of the prepared cationic surfactants was elucidated via IR, ^1^H, and ^13^C NMR spectra.

ICS-10: Yellow, solid color, mp= 105 °C, and yield = 93%. FT-IR (KBr pellet) cm^−1^: 3382, 3013, 2952, 2923,1749, 1653, 1601, 1539, and 1252. ^1^H NMR-400 MHz (DMSO-*d*_6_) d ppm: 0.84(t, 3H, CH_3_-CH_2_−), 1.25(s, 14H, (CH_2_)7−), 1.64(m, 2H, CH_2_-CH_2_-O), 2.15(m, 2H, CH_2_CH_2_-N), 3.64(t, 2H, CH_2_-N+), 4.13(s, 6H, 2CH_3_-N+), 4.20(t, 2H, CH_2_-N=CH), 4.53(t, 2H, CH_2_-O), 4.56(t, 2H, CH2-CO), 6.94 (d, 1H, aromatic CH), 7.34(m, 2H, aromatic CH), 7.44(m, 1H, aromatic CH), and 8.60(s, 1H, CH=N). ^13^C NMR-400 MHz, δC (ppm) (DMSO-d6): 167, 165, 161, 132, 131, 118, 115, 66, 63, 61, 55, 51, 31, 29.46, 29.17, 29.06, 28, 25, 24, 22, and 14.01. (Appendix A).

ICS-14: Pale-yellow solid color, mp= 110 °C, and yield= 95%. FT-IR (KBr pellet) cm^−1^: 3268, 3063, 2919, 2850,1751, 1642, 1611, 1537, 1445, and 1254. ^1^H NMR-400 MHz (DMSO-d6) d ppm: 0.84(t, 3H, CH_3_-CH_2_−), 1.25(s, 20H, (CH_2_)_10_−), 1.63(m, 2H, CH_2_-CH_2_-O), 1.85(m, 2H, CH_2_CH_2_-N), 1.99(t, 2H, CH_2_-N+), 3.01(s, 6H, 2CH_3_-N+), 3.25(t, 2H, CH_2_-N=CH), 3.81(t, 2H, CH_2_ CH_2_-O), 4.24(t, 2H, CH_2_-O), 4.60(t, 2H, CH_2_-CO), 7.03(m, 1H, aromatic CH), 7.42 (s, 2H, aromatic CH), and 7.53(m, 1H, aromatic CH), 8.56(s, 1H, CH=N). ^13^C NMR (151 MHz, DMSO-*d*_6_) δ: 166.41, 164.67, 161.18, 144.56, 134.05, 127.79, 118.18, 115.52, 68.23, 63.64, 60.96, 54.27, 50.65, 31.76, 29.47, 29.41, 29.30, 29.17, 28.97, 26.22, 22.94, 22.56, 22.11, and 14.43 (Figure 8A–C).

### 3.4. Weight Loss (WL) Measurements

In the weight loss (WL) method, C-steel coupons with dimensions of 4.8 × 2 × 1.1 cm with an uncovered surface area of 34.16 cm^2^ were used. These coupons were mechanically brushed at Benha University with various grades of emery paper before being rinsed with distilled water, propanone, and distilled water. Before adding C-steel samples into the test solution, the weight of the samples was determined. The experimentation was replicated 3 times, with the average WL reported each time. The following equations were utilized to calculate the surface coverage (*θ*) and the inhibition capacity (*IE*/%):(11)$$IE/\%=\left(1-\frac{\Delta W_{surf}^{}}{\Delta W_{free}^{}}\right)\times 100=\theta \times 100$$
where Δ*W_surf_* and Δ*W_free_* reflect the weight change of C-steel samples per unit area with and without ICS-10 and ICS-14 surfactants, respectively. The next equation was used to measure the corrosion rate (*CR*) in g·hr^−1^.cm^−2^:(12)$$CR=\frac{\Delta W}{t\times A}$$
where Δ*W* signifies the change in the weight of the specimen in grams, *A* is the sample area in cm^2^, and t denotes the time spent immersed in hours.

### 3.5. Electrochemical Measurements

Because corrosion occurs via electrochemical reactions, electrochemical techniques are ideal for the study of corrosion processes [65,66]. Electrochemical measurements were performed by a Meinsberg Potentiostat/Galvanostat electrochemical workstation (PS6, Hamburg, Germany), complemented with the EC-Lab software PS Remote, at Benha University. The Galvanostat/Potentiostat was committed in another way to a conventional cell with three-electrode systems. The counter and reference electrodes were a Pt-wire and a saturated calomel electrode (SCE), respectively. The working electrode was made of a C-steel that was encased in Araldite resin, with just 1.6 cm^2^ of the C-steel electrode apparent. The C-steel was rubbed with emery papers of up to a 2500 grade before being degreased with propanone and washed with distilled water. The potentiodynamic polarization (PDP) plots for C-steel corrosion in 1.0 M of HCl in the absence and presence of various amounts of ICS-10 and ICS-14 surfactants were determined. All experiments were conducted with a scan rate of 1.0 mV/s [67,68,69].

### 3.6. SEM Observations

The SEM examination was carried out at Mansoura University in Egypt [70] with the help of the JSM-6510LV. The C-steel specimen was scratched with emery papers of up to a 2500 grade and then maintained in 1.0 M of HCl for 24 h in the absence of 0.0005 M of ICS-10 and ICS-14 surfactants. After this period of inundation, the samples were washed with distilled H_2_O, properly desiccated, and placed in the spectrometer.

### 3.7. Computational Details

Full geometry optimization of the two inhibitors’ ICS10 and ICS14 molecules were studied without any constraints using DFT calculations with B3LYP functional [71,72,73] and 6–31 g(d,p) basis sets implemented in the GAMESS program [74,75], and the calculations were carried out in the gas phase. The results were evaluated to include the highest occupied molecular orbital (HOMO), the energy gap (Δ*E*), the lowest unoccupied molecular orbital (LUMO), the dipole moment (*µ*), hardness (*η*), global softness (*σ*), and the number of electrons transferred (Δ*N*). The *η*, *σ*, and Δ*N* were computed by:(13)$$\eta =\frac{E_{LUMO}-E_{HOMO}}{2}$$
(14)$$\sigma =\frac{1}{\eta }$$
(15)$$\Delta N=\frac{\phi -\chi_{inh}}{2(\eta_{Fe}+\eta_{inh})}$$
where $\varphi ,\chi_{inh}$, $\eta_{Fe},and\eta_{inh}$ are the function work of Fe (110) (4.820 eV), the surfactant electronegativity, iron chemical hardness (0 eV), and surfactant chemical hardness, respectively. 

### 3.8. MC Simulations

In this study, an Fe (110) surface was selected for the adsorption of a single inhibitor molecule. MC simulation was performed by the adsorption locator module. The Fe (110) surface was increased by constructing a supercell, 15 × 15, with 25 Å vacuum heights above the surface. Condensed-phase optimized molecular potentials for atomistic simulation studies (COMPASS) were utilized for the adsorption process. For the summation method, the Ewald method was set for the electrostatic and atom-based method for van der Waals. Additional information on the methodology of Monte Carlo simulations can be found in previous works [76]. The adsorption energy of the inhibitor was calculated by the following equation:*E*_ads_ = *E*_complex_ − *E*_Fe_ − *E*_inh_
(16)
where *E*_complex_, *E*_Fe_, and *E*_inh_ are the total energies of the inhibitor on the Fe (110) surface, the energy of the iron (110) surface, and the energy of the surfactant molecule, respectively.

## 4. Conclusions

Two cationic imine surfactants were synthesized, and we elucidated their chemical structures, which showed good surface-active properties. These prepared imine cationic surfactants (ICS-10 and ICS-14) exposed the capability of producing a defensive layer against the aggressive acidic medium at diverse temperatures via the adsorption of their molecules on the metal surface. Surface inspection through SEM displayed a significant inhibition of C-steel deterioration by the construction of a protecting film on the metal interface. Interestingly, the small negative values suggest that the adsorption process is spontaneous. There was no considerable shift in the potential corrosion values, inferring that these imine compounds act as mixed-type inhibitors. Theoretical calculations based on DFT are in agreement with the experimental findings that exhibited ICS-10 and ICS-14 as efficient corrosion additives, and the order of the inhibition capability for the investigated imine surfactants is as follows: ICS-14 (94.58%) > ICS-10 (91.53 %).

## Data Availability

The raw/processed data generated in this work are available upon request from the corresponding author.

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
