# Peer review of "Novel Imine-Tethering Cationic Surfactants: Synthesis, Surface Activity, and Investigation of the Corrosion Mitigation Impact on Carbon Steel in Acidic Chloride Medium via Various Techniques"

_molecules, 2023, doi:10.3390/molecules28114540_

Round 1
Reviewer 1 Report
Reviewer Comment
Manuscript Number: molecules-2411157
Journal : Molecules
Dear Editor,
This paper deals the “Novel synthesized Imines Surfactants: Synthesis, surface activity, and investigation of the Corrosion mitigation impact on Carbon steel in acidic chloride medium by chemical, electrochemical, and theoretical techniques” . The manuscript is very interesting. Moreover, I think the results should be of interest to the readership of “Molecules”. However, I recommend reconsideration of the manuscript after major Revision. I would suggest acceptance after revision.
1. The title can be better and more attractive; it should be more precise and represent to the contents
2. Qualitative informations are missing in abstract. Abstract should be concise and the authors need to improve with more specific short results.
3. The novelty of the work has not been demonstrated appropriately in the introduction section.
4. In the introduction section, the number of recent references (2016 – 2022) is limited. The following references should be included in the manuscript.
- El Kacimi, Y., R. Touir, M. Galai, R. A. Belakhmima, A. Zarrouk, K. Alaoui, M. Harcharras, H. El Kafssaoui, and M. Ebn Touhami. "Effect of silicon and phosphorus contents in steel on its corrosion inhibition in 5 M HCl solution in the presence of Cetyltrimethylammonium/KI." J Mater Environ Sci 7, no. 1 (2016): 371-381.
-Dahmani, K., et al. "Quantum chemical and molecular dynamic simulation studies for the identification of the extracted cinnamon essential oil constituent responsible for copper corrosion inhibition in acidified 3.0 wt% NaCl medium." Inorganic Chemistry Communications 124 (2021): 108409.
-Rbaa, M., M. Galai, Ashraf S. Abousalem, B. Lakhrissi, M. Ebn Touhami, I. Warad, and Abdelkader Zarrouk. "Synthetic, spectroscopic characterization, empirical and theoretical investigations on the corrosion inhibition characteristics of mild steel in molar hydrochloric acid by three novel 8-hydroxyquinoline derivatives." Ionics 26 (2020): 503-522.
5. All figure captions should provide with all the necessary information.
6. The discussion part of Potentiodynamic polarisation (PDP) studies is very poor. It should be improved.
7. Comparative analysis of the present data with those published in the literature for the similar type of inhibitor would support and can improve the quality of discussion
8. Mechanism of the inhibition is poor, and it is an important part for better understanding, for instance, in the first step on the electrode (in the corrosion or inhibition processes); what it is happening, then what it is follows and so on. At least, three or four steps are happing before the inhibitor's adsorption on the metal surface.
9. In the Adsorption studies section, can you use other isotherm models to confirm the first one
Author Response
Author's response
< Molecules Ms. No.: molecules-2411157>
< Novel synthesized Imines Surfactants: Synthesis, surface activity, and investigation of the Corrosion mitigation impact on Carbon steel in acidic chloride medium by chemical, electrochemical, and theoretical techniques>
Dear Professor,
We, the authors, would like to thank you for giving us a chance to resubmit the paper and also thank the editor for giving us constructive suggestions which would help us to improve the quality of the paper. Here, we submit the revised version of our manuscript, which has been modified according to the editor. All changes and modifications were highlighted with yellow color in the new version manuscript. The detailed corrections are listed below point by point:
We would like to thank the reviewer for their great efforts and for giving useful criticism of the article. Below are the answers to each point.
Reviewer #1:
This paper deals with the “Novel synthesized Imines Surfactants: Synthesis, surface activity, and investigation of the Corrosion mitigation impact on Carbon steel in acidic chloride medium by chemical, electrochemical, and theoretical techniques”. The manuscript is very interesting. Moreover, I think the results should be of interest to the readership of “Molecules”. However, I recommend reconsideration of the manuscript after Major Revision. I would suggest acceptance after revision.
1-The title can be better and more attractive; it should be more precise and represent to the contents.
Author reply:
First, thanks for pointing out this. The title was modified as shown below: (Novel Imine-tethering cationic Surfactants: Synthesis, surface activity, and investigation of the Corrosion mitigation impact on Carbon steel in acidic chloride medium via various techniques)
2-Qualitative informations are missing in abstract. Abstract should be concise and the authors need to improve with more specific short results.
Author reply:
Done and modified
3-The novelty of the work has not been demonstrated appropriately in the introduction section.
Author reply:
Thank you for your kind comment. Cationic surfactants bearing Schiff-base moiety proved high efficiency in an acidic solution, where a nitrogen atom is accomplished to form coordinate-covalent bonds with the steel interface by its unshared lone-pairs of electrons; Whereas the π-bond interacts physically increasing their adsorption at-traction to the metal interface. The imine surfactant molecules are categorized by their high capability to adsorb at the interfaces owing to their unique amphipathic construction, which contains two opposing parts tail and head. The head is considered by its high electronic rich, which is about a set of some electronic rich functional groups like double, carbonyl, nitrogen, imine group, oxygen, etc. A successive film on the metal surface could be shaped by the surfactant hydrocarbon which has the capability to separate the corroded metal surface from contact with corrosive mediums [29-34]. In addition, the imine cationic surfactant is considered by its biocidal activity which improves its capacity in the petroleum sector. (Please see the introduction part, lines 62-78).
4-In the introduction section, the number of recent references (2016 – 2022) is limited. The following references should be included in the manuscript.
- El Kacimi, Y., R. Touir, M. Galai, R. A. Belakhmima, A. Zarrouk, K. Alaoui, M. Harcharras, H. El Kafssaoui, and M. Ebn Touhami. "Effect of silicon and phosphorus contents in steel on its corrosion inhibition in 5 M HCl solution in the presence of Cetyltrimethylammonium/KI." J Mater Environ Sci7, no. 1 (2016): 371-381.
- Dahmani, K., et al. "Quantum chemical and molecular dynamic simulation studies for the identification of the extracted cinnamon essential oil constituent responsible for copper corrosion inhibition in acidified 3.0 wt% NaCl medium." Inorganic Chemistry Communications124 (2021): 108409.
- Rbaa, M., M. Galai, Ashraf S. Abousalem, B. Lakhrissi, M. Ebn Touhami, I. Warad, and Abdelkader Zarrouk. "Synthetic, spectroscopic characterization, empirical and theoretical investigations on the corrosion inhibition characteristics of mild steel in molar hydrochloric acid by three novel 8-hydroxyquinoline derivatives." Ionics26 (2020): 503-522.
Author reply:
Done and added
5- All figure captions should provide with all the necessary information.
Author reply:
Done and modified
6- The discussion part of Potentiodynamic polarisation (PDP) studies is very poor. It should be improved.
Author reply:
The Potentiodynamic polarisation section has been improved. (see 3.4. Potentiodynamic polarization (PDP))
7-Comparative analysis of the present data with those published in the literature for the similar type of inhibitor would support and can improve the quality of discussion
Author reply:
Thanks a lot for your comment. As shown in Table S3, the inhibition powers of our synthesized Imines Surfactants (~ 95) are close to or slightly higher than that of the previous synthetic Schiff base compounds (~94) [Negm et al., 2009; Fouda et al., 2012; Abd El-Lateef & Tantawy, 2016; Hami-touche et al., 2013]. Thus, in spite of the yield products accounts, the use of the synthesized compounds like our synthesized cationic surfactants for protection of the metal surfaces in the corrosive solutions containing chloride is more yield than previous synthetic compounds containing hydroxyl group [Abd El-Lateef & Tantawy, 2016].
8- Mechanism of the inhibition is poor, and it is an important part for better understanding, for instance, in the first step on the electrode (in the corrosion or inhibition processes); what it is happening, then what it is follows and so on. At least, three or four steps are happing before the inhibitor's adsorption on the metal surface.
Author reply:
The Mechanism section has been improved. (3.9. MC simulation and the mechanism of inhibition)
9-In the Adsorption studies section, can you use other isotherm models to confirm the first one
Author reply:
Different models of adsorption isotherm, for example, the Langmuir, Temkin, Freundlich, and Frumkin models, were utilized to fit the weight loss data to comprehend how the surfactant compounds are adsorbed on the steel interface. The closest match was for the Langmuir adsorption isotherm. (3.6. Adsorption isotherm considerations)
The Temkin, Freundlich, and Frumkin isotherms is given below:

Reviewer 2 Report
The author needs to include the appropriate working principal figure in the introduction for better understanding.
The bibliography is very old and is not in accordance with the journal.
The quality the figures 3 and 4 are poor and should be improved English should be improved by further proofreading the paper and removing the typos and grammatical errors from the text. There are also some complex sentences that do not read well.
Explain more the DFT and MC calculation methods.
Language of the manuscript should be checked in detail. It is far from academic writing language. Get help from professionals or natives if necessary.
Author Response
Author's response
< Molecules Ms. No.: molecules-2411157>
< Novel synthesized Imines Surfactants: Synthesis, surface activity, and investigation of the Corrosion mitigation impact on Carbon steel in acidic chloride medium by chemical, electrochemical, and theoretical techniques>
Dear Professor,
We, the authors, would like to thank you for giving us a chance to resubmit the paper and also thank the editor for giving us constructive suggestions which would help us to improve the quality of the paper. Here, we submit the revised version of our manuscript, which has been modified according to the editor. All changes and modifications were highlighted with yellow color in the new version manuscript. The detailed corrections are listed below point by point:
We would like to thank the reviewer for their great efforts and for giving useful criticism of the article. Below are the answers to each point.
Reviewer #2:
1-The author needs to include the appropriate working principal figure in the introduction for better understanding.
Author reply:
Done and modified
2-The bibliography is very old and is not in accordance with the journal.
Author reply:
Done and modified
3-The quality the Figures 3 and 4 is poor and should be improved English should be improved by further proofreading the paper and removing the typos and grammatical errors from the text. There are also some complex sentences that do not read well.
Author reply:
Thank you very much for giving us the opportunity to revise our manuscript.
We have made corrections to the grammar and English usage with the help of my native English-speaking teacher, and this revision can make our paper more acceptable. Moreover, based on the reviewer’s suggestion; we have carefully revised the manuscript by use of “premium Grammarly (https://app.grammarly.com/ddocs/427154829)”. The revised details can be found in the revised version.
4-Explain more the DFT and MC calculation methods.
Author reply:
Done and modified

Reviewer 3 Report
This work is interesting and it also combines the experimental data and the theoretical results perfectly. I recommend the publication of this work in Molecules after minor revision. A few comments can be found as follows:
1. How did you calculate the inhibition capacity (%) in Fig.2, please add the corresponding formula in the manuscript.
2. What’s the difference and correlation between section 2.6 and 2.7? If the results of them are in good agreement, you should emphasize this point to the readers.
3. Is there any correlation between DFT calculations (section 2.8) and MC simulation (section 2.9)? Does the DFT calculation could provide any instruction significance for the MC simulation?
English can be further improved.
Author Response
Author's response
< Molecules Ms. No.: molecules-2411157>
< Novel synthesized Imines Surfactants: Synthesis, surface activity, and investigation of the Corrosion mitigation impact on Carbon steel in acidic chloride medium by chemical, electrochemical, and theoretical techniques>
Dear Professor,
We, the authors, would like to thank you for giving us a chance to resubmit the paper and also thank the editor for giving us constructive suggestions which would help us to improve the quality of the paper. Here, we submit the revised version of our manuscript, which has been modified according to the editor. All changes and modifications were highlighted with yellow color in the new version manuscript. The detailed corrections are listed below point by point:
We would like to thank the reviewer for their great efforts and for giving useful criticism of the article. Below are the answers to each point.
Reviewer #3:
This work is interesting and it also combines the experimental data and the theoretical results perfectly. I recommend the publication of this work in Molecules after minor revision. A few comments can be found as follows.
1- 1. How did you calculate the inhibition capacity (%) in Fig.2, please add the corresponding formula in the manuscript.
Author reply:
Thank you for your comment. The following equations were utilized to calculate the surface coverage (θ) and the inhibition capacity (IE/%):
(1)
2-2. What’s the difference and correlation between sections 2.6 and 2.7? If the results of them are in good agreement, you should emphasize this point to the readers
Author reply:
There is no correlation between sections 2.6 and 2.7
3- Is there any correlation between DFT calculations (section 2.8) and MC simulation (section 2.9)? Could the DFT calculation provide any instruction significance for the MC simulation?.
Author reply:
DFT study provides the inhibitor properties through some quantum parameters such as energy gap. MC simulation provides the adsorption of inhibitors on the steel surface. From the DFT study, the energy gap of the ICS-14 inhibitor is smaller than ICS-10, which gives an indication of easier adsorption on the mild steel.
I hope that all modifications have taken into account the suggestions of the editor and reviewers. The manuscript has been resubmitted to your journal. We look forward to your positive response.

Reviewer 4 Report
Synthesized of imines surfactants is certainly an interesting topic. The description of the experiments is complete, experimental results are evaluated; discussions are supported by the experimental results, but I have some questions and observations for authors.
1. Line 98. Please write, what means CMC (critical micelle concentration) and after use the abbreviations.
2. Line 120. 25℃
3. The authors do not mention in text Figure 1B.
4. The references from [32] to [43] do not appear in text.
5. Line 131: reference for equation (7) missing.
6. Line 210: reference for equation (12) missing.
7. Line 333-334: references [35, 36] appears in text after reference [60] and do not mention before.
8. The reference [32] appear at line 435. Please reorder the references.
Author Response
Author's response
< Molecules Ms. No.: molecules-2411157>
< Novel synthesized Imines Surfactants: Synthesis, surface activity, and investigation of the Corrosion mitigation impact on Carbon steel in acidic chloride medium by chemical, electrochemical, and theoretical techniques>
Dear Professor,
We, the authors, would like to thank you for giving us a chance to resubmit the paper and also thank the editor for giving us constructive suggestions which would help us to improve the quality of the paper. Here, we submit the revised version of our manuscript, which has been modified according to the editor. All changes and modifications were highlighted with yellow color in the new version manuscript. The detailed corrections are listed below point by point:
We would like to thank the reviewer for their great efforts and for giving useful criticism of the article. Below are the answers to each point.
Reviewer #4:
Synthesizing of imine surfactants is certainly an interesting topic. The description of the experiments is complete, experimental results are evaluated; discussions are supported by the experimental results, but I have some questions and observations for authors.
1-Line 98. Please write, what means CMC (critical micelle concentration) and use the abbreviations.
Author reply:
Done (line 227)
2-Line 120. 25℃
Author reply:
Done
3-The authors do not mention in text Figure 1B.
Author reply:
Mentioned and added
4-The references from [32] to [43] do not appear in text.
Author reply:
Done
5-Line 131: reference for equation (7) missing.
Author reply:
Done and Added
6-Line 210: reference for equation (12) missing.
Author reply:
Done and Added
7-Line 333-334: references [35, 36] appears in text after reference [60] and do not mention before.
Author reply:
Corrected and added
8-The reference [32] appear at line 435. Please reorder the references.
Author reply:
Reordered
I hope that all modifications have taken into account the suggestions of the editor and reviewers. The manuscript has been resubmitted to your journal. We look forward to your positive response.

Round 2
Reviewer 2 Report
I agree with the publication of the paper.